# COVID-19 Pandemic and Consumption of Dietary Supplements among Adult Residents of Lithuania

**DOI:** 10.3390/ijerph19159591

**Published:** 2022-08-04

**Authors:** Rokas Arlauskas, Donatas Austys, Rimantas Stukas

**Affiliations:** Department of Public Health, Institute of Health Sciences, Faculty of Medicine, Vilnius University, M. K. Čiurlionio 21/27, LT-03101 Vilnius, Lithuania

**Keywords:** dietary supplements, adults, Lithuania, COVID-19

## Abstract

In the light of the changing pattern of the consumption of dietary supplements among adults in other countries during the COVID-19 pandemic, we aim to assess the prevalence of the consumption of dietary supplements with respect to the purpose of consumption and sociodemographic factors among adults in Lithuania. This study included two samples of adults from Lithuania: 870 in 2019 and 1600 in 2021. Anonymous original questionnaires were used to collect the data about the consumption of dietary supplements before and during the pandemic. The consumption of dietary supplements was prevalent among 67.2% and 78.1% of the samples, respectively. Although the majority (73.7%) of the respondents indicated that the pandemic had no effect on their use of dietary supplements, every fourth respondent’s (24.6%) consumption increased. In 2021, the consumption was more prevalent among females, respondents with university education, urban citizens, employed respondents, respondents without children, with higher income, and those who knew a person with COVID-19 (*p* < 0.05). During the pandemic, the strengthening of the immune system and the body were the leading reasons for consumption (49.0% and 43.5%, respectively). The consumption of dietary supplements appropriate for these purposes increased among 63.3% and 66.9% of respondents, respectively. To conclude, the consumption of dietary supplements among adults in Lithuania increased during the COVID-19 pandemic.

## 1. Introduction

Studies show that the nutritional habits of a large part of adult population are not favorable for health [1,2]. As recommended by WHO and national health care institutions, each day people should obtain a wide variety of nutrients [3,4]. According to the European Food Safety Authority, dietary supplements are intended to correct nutritional deficiencies, maintain an adequate intake of certain nutrients, or support specific physiological functions [5].

According to the results of other surveys, the COVID-19 pandemic had negative and positive effects on the nutrition and lifestyle of the majority of adults. A reduction in physical activity and an increase in the consumption of alcohol, snacks and sweets, but also fruits, vegetables, nuts, legumes, and fish or shellfish were reported [6,7,8,9]. During the pandemic, an increase in the frequency of searching for information on improving the immune system by food products or dietary supplements was observed [10,11]. Dietary supplements became the field of interest of scientists as well. The possible use of dietary supplements as a likely measure to stop the spread of SARS-CoV-2 (2019-nCoV) was analyzed. The immune-boosting, antiviral, antioxidant, or anti-inflammatory effects of dietary supplements were emphasized [12]. During the pandemic, sales of dietary supplements dramatically rose despite the depressed economic conditions [13]. A study conducted in Poland showed that the consumption of dietary supplements containing zinc and vitamin D increased during the pandemic, especially among people with higher education, or those with medical and paramedical education. With respect to the nonrestrictive registration procedures of dietary supplements, the authors emphasized the necessity of educating consumers in terms of the selection of appropriate preparations, proper nutrition, and balanced supplementation [6]. However, there is a lack of representative studies about the prevalence of consumption of various dietary supplements in various sociodemographic groups in different countries. Taking into account that there are no studies about the prevalence of consumption of dietary supplements during the pandemic in Lithuania, the aim of our study is to assess the prevalence of the consumption of dietary supplements focusing on the purpose of the consumption among social and demographic groups in Lithuania during the COVID-19 pandemic in comparison with the pre-pandemic period.

## 2. Materials and Methods

### 2.1. Procedure of Data Collection

The main data for this study were collected during the COVID-19 pandemic, in October and November of 2021. A representative sample of 1600 adults aged from 18 to 64 years was gathered. Male and female residents of Lithuania were included. A multistage stratified probabilistic sampling method was used to select the participants for this study from the registry of the residents of Lithuania. It ensured an equal probability for every household in the country to be surveyed and, according to the target criteria, the sample represented the general population. Data were collected by conducting computer-assisted web interviews (CAWIs). Every selected resident received an invitation to participate in this study with a link to the anonymous questionnaire by email. The participants of this study filled in the questionnaire by themselves at a time convenient to them. It was possible to fill the questionnaire only once.

For the purpose of comparison with the pre-pandemic period, an additional data sample of 905 adults aged from 18 to 64 years was included. It included data collected in March 2019. The selection of the participants was analogous to the one used in 2021. However, 60% of the respondents were interviewed by means of computer-assisted personal interviews (CAPIs) and 40% of the respondents were interviewed by means of CAWIs. CAPIs were performed by a professional interviewer using a questionnaire prepared in advance. After the exclusion of 35 (3.9%) respondents who were unable to answer the question on whether they had consumed dietary supplements within the past 12 months, we concluded with a sample of 870 respondents whose answers were used for statistical analysis.

In cases of refusal to participate in the study by invited individuals, additional participants were randomly selected and invited from a group of people of the same age, sex, and geographical characteristics.

The distribution of the respondents in both samples by social and demographic factors is presented in Table 1.

This study was reviewed by the Vilnius Regional Ethics Committee for Biomedical Research. The participants provided their consent by filling out the anonymous questionnaires.

### 2.2. Description of the Questionnaire

Two anonymous validated original questionnaires were used for this study: one in 2019 and the other in 2021. We used a part of the first questionnaire. It included one question about the consumption of dietary supplements, “What dietary supplements and what for have you taken over the last 12 months?”, with the answer options, “For strengthening the immune system/For disease prevention and the overall strengthening of the body/For energy boosting/For eye care/For boosting memory/For boosting the nervous system/For strengthening the cardiovascular system/For strengthening the joints, bones/For better digestion/For sleep regulation/For athletes/For weight regulation/I have not taken any dietary supplements within the last 12 months/I cannot answer”. In addition, 5 questions about social and demographic characteristics were included.

In 2021, the questionnaire included questions about the social and demographic characteristics of the respondents, COVID-19 experience, subjective assessment of personal health, nutrition, consumption of dietary supplements, and physical activity. The questionnaire was constructed on the basis of the questionnaire used previously in 2019 [14], by adding additional questions related to the COVID-19 pandemic and its possible impact on nutrition, consumption of dietary supplements, and physical activity. In this paper, we present the analysis of part of the questions included in the questionnaire (Table 2).

The questionnaire also included a group of questions about social and demographic factors. Two of those questions regarding the respondents’ age and place of residence were open-ended. To achieve an unambiguous interpretation of the results, we transformed them into a binary format. Respondents were asked to identify the municipality they live in. Respondents from 5 municipalities with the largest number of residents were assigned to the “City” group, while all the rest respondents were attributed to the “Towns and villages” group. The age was categorized by a median to the range up to 41-year-olds and from 42-year-olds. All other questions were close-ended. Respondents with primary or secondary education and those who finished a high school were assigned to the “Non-university education” group. Respondents with unfinished or finished university studies were assigned to the “University education” group. In terms of the employment status, the “Employed” and “Unemployed” groups were created. Heads of companies or the departments, office workers, civil servants, service sector employees, sellers, workers, and farmers were assigned to the “Employed” group. Retirees, housewives, persons on parental leave, non-employed persons, and students were categorized as “Unemployed”. In addition to this, more binary variables were created, such as marital status, with children up to 18 years old, and the income of a family per member. The categorization of the rest of the questionnaire is presented in Table 2.

### 2.3. Statistical Analysis

In order to determine the change in the consumption of different dietary supplements during the pandemic, the respondents were asked to indicate which of the dietary supplements they were made to consume more under the impact of the COVID-19 pandemic. A list of 20 dietary supplements was provided (Table 2). During the statistical analysis, every dietary supplement from this list was assigned to one or more categories with an emphasis on the purpose of the consumption of those dietary supplements. The consumption of at least one dietary supplement in a category was defined as an increase in the consumption of dietary supplements for a specific purpose during the pandemic (Table 3). This method served us as a tool for the unambiguous comparison of consumers’ expectations of the consumption of dietary supplements and the human body parts affected by the consumption of exact dietary supplements.

The normality of distribution of variables was tested using the Shapiro–Wilk test. With respect to the results of this test, medians with interquartile range (Q1–Q3) were presented for variables with non-normal distribution, and averages with standard distributions were presented for variables with normal distribution. Pearson’s chi-squared test (χ2) was used to determine whether there was a statistically significant difference between the expected frequencies and the observed frequencies in one or more categories. Differences were considered statistically significant when the *p*-value was lower than 0.05.

## 3. Results

### 3.1. Characteristics of the Samples

In 2021, the median age of the respondents was 42 (29–54) years. The majority of the respondents were employed, married (or had partners), from small towns or villages, people with university education, without children up to 18 years old. In 2019, the median age of the respondents was 42 (30–52) years. This sample included relatively more employed, single, and individuals with university education (*p* < 0.05). In both samples, the distribution of the respondents was similar according to sex, age, and place of residence (*p* > 0.05) (Table 1).

### 3.2. The Prevalence of the Consumption of Dietary Supplements in Both Samples

In 2019, the consumption of dietary supplements was prevalent among 66.1% of the respondents.

In 2021, the majority (78.1%) of the respondents indicated the consumption of dietary supplements: everyday consumption—24.0%, 6-month consumption—9.8%, consumption from 4 to 6 months—10.3%, consumption from 2 to 3 months—16.2%, one-month consumption—4.9%, short or accidental consumption—12.9%, and no consumption—21.9%.

As presented in Figure 1, the consumption of dietary supplements significantly increased among males, married, and unemployed persons (*p* < 0.05).

### 3.3. Consumption of Dietary Supplements during the COVID-19 Pandemic

In 2021, vitamin C, vitamin D, omega-3 fatty acids, magnesium, and dietary supplements for the general strengthening of the body were among the most frequently used dietary supplements highlighted in response to the question which of the dietary supplements were they made to consume more under the impact of the COVID-19 pandemic (Table 4).

During the pandemic, the consumption of dietary supplements was more prevalent among females, people with university education, respondents from cities, employed respondents, those without children, and with higher income. Additionally, those who suffered from COVID-19 or it occurred to the respondents’ family members or friends consumed dietary supplements more frequently. The distribution of the respondents by their answers to the question if they consumed dietary supplements was similar in different groups of age, COVID-19 cases in respondents’ families, and the degree of the severeness of COVID-19 (*p* > 0.05) (Table 5).

Almost a quarter (26.1%) of the respondents indicated that the COVID-19 pandemic aroused their interest in dietary supplements. An increased interest was prevalent among a larger part of older respondents and those who had suffered from COVID-19 or their family members did. Other factors such as sex, education, place of residence, marital status, employment, income, having children, and the degree of the severeness of COVID-19 were not associated with an increase in the interest in supplements because of the pandemic (Table 5).

Suffering from COVID-19 disease personally or by family members or friends was associated with an increase in the consumption of dietary supplements during the pandemic. Other factors such as sex, age, education, place of residence, marital status, children in the family, employment, income, or COVID-19 severeness were not associated with an increase in the consumption of dietary supplements during the pandemic (Table 5).

### 3.4. The Consumption of Dietary Supplements within the Past 12 Months and Increase in the Consumption during the Pandemic by Social and Demographic Groups

As presented in Table 6, in comparison with the 2019 data, a significant increase in the consumption of dietary supplements for strengthening the immune system, the cardiovascular system, the joints, and bones was observed during the pandemic (*p* < 0.05). Additionally, the analysis of data revealed a significant decrease in the consumption of dietary supplements for boosting energy, memory, eye care, regulation of sleep, and athletics (*p* < 0.05).

In 2021, the largest part of the respondents indicated that they were consuming dietary supplements within the past 12 months in order to strengthen their immune system, also in order to generally strengthen their bodies. Similarly, after the analysis of the consumption of the exact dietary supplements, these purposes were also the most prevalent. On the other hand, the differences were found in the prevalence of the consumption of dietary supplements mainly impacting other systems of the human body. In most cases, the difference was favorable; however, in the case of consumption of dietary supplements to improve digestion, it was six times lower (Table 6).

During the pandemic, the consumption of all the mentioned dietary supplements within the past 12 months was similar among males and females, except the consumption of dietary supplements for vision. The consumption of such dietary supplements was indicated by 64 (10.8%) males and 99 (15.3%) females (*p* = 0.021). However, the analysis of the indicated increase in the consumption of the exact dietary supplements was similar among males and females for all the mentioned purposes (*p* > 0.05) (Table 6).

In 2021, younger respondents more frequently consumed dietary supplements for strengthening the immune system, boosting energy, the nervous system, for regulation of sleep, and also for athletics (respectively, 53.2% vs. 45.4%, 18.2% vs. 9.9%, 27.2% vs. 20%, 12.2% vs. 7.6%; *p* < 0.05). Older respondents more frequently consumed dietary supplements for strengthening their cardiovascular system, and also for joints and bones (respectively, 15.8% vs. 36.7% and 18.8% vs. 30.7%; *p* < 0.05). The consumption of dietary supplements for the general strengthening of the body, vision, memory, digestion, and COVID-19 prevention was similar among both age groups (*p* > 0.05). An increase in the consumption of dietary supplements was more common among younger respondents in the case of dietary supplements for strengthening the immune system and the general strengthening of the body (respectively, 58.9% vs. 67.2% and 62.3% vs. 71%; *p* < 0.05). An increase in dietary supplements for the cardiovascular system and also for joints and bones was more frequently observed among older respondents (respectively, 31.5% vs. 38.2% and 44.8% vs. 50.7%; *p* < 0.05). An increase in the consumption of dietary supplements appropriate for boosting energy, vision, memory increase, nervous system, digestion, regulation of sleep, and athletics was similar among both, younger, and older respondents (*p* > 0.05) (Table 6).

During the pandemic, the consumption of dietary supplements for the nervous system and for the regulation of sleep was more frequent among the respondents with non-university education (respectively, 28% vs. 21.5% and 14% vs. 8.1%; *p* < 0.05). The consumption of dietary supplements for other purposes was similar among the respondents in the higher and lower education groups (*p* > 0.05). An increase in the consumption of dietary supplements appropriate for boosting energy, vision, memory, and athletics was also more pronounced among the respondents with non-university education (respectively, 26.9% vs. 20.4%, 25.4% vs. 19.7%, 34.3% vs. 27.8% and 26.9% vs. 20.4%; *p* < 0.05). An increase in the exact dietary supplements for other purposes was not associated with education (*p* > 0.05).

In 2021, respondents from small towns or villages more frequently indicated the consumption of dietary supplements for improving digestion (13.1% vs. 18.8%; *p* = 0.007). Respondents from cities more frequently consumed dietary supplements for athletics (7.8% vs. 3.8%, *p* = 0.002). The consumption of dietary supplements for other purposes was similar among residents from cities and towns or villages (*p* > 0.05). An increase in the consumption of the exact dietary supplements for all mentioned purposes was similar among residents from cities and towns or villages (*p* > 0.05) (Table 6).

During the pandemic, the consumption of dietary supplements for boosting energy, the nervous system, and athletics was more frequent among single (unmarried and living without a partner) respondents (respectively, 18.1% vs. 12.7%, 28.9% vs. 22.1% and 8.8% vs. 4.9%; *p* < 0.05). The consumption of dietary supplements for the cardiovascular system was more prevalent among married respondents (20.7% vs. 28.4%; *p* = 0.011). The consumption of dietary supplements for other purposes was not related to the marital status (*p* > 0.05). Additionally, an increase in the consumption of the exact dietary supplements for all the mentioned purposes was similar among married and single respondents (*p* > 0.05) (Table 6).

During the pandemic, the consumption of dietary supplements for strengthening the immune system was more prevalent among respondents with children under 18 years old (46.8% vs. 52.9%; *p* = 0.04). The consumption of dietary supplements for strengthening the cardiovascular system and bones and joints was more prevalent among the respondents with no children under 18 years old (respectively, 30.7% vs. 20.6% and 28.2% vs. 19.5%; *p* < 0.05). The consumption of dietary supplements for all other purposes was not associated with having children under 18 years old (*p* > 0.05). An increase in the consumption of dietary supplements appropriate for boosting memory was more frequent among the respondents with children under 18 years old (32.4% vs. 26.9%; *p* = 0.042); the consumption of the exact dietary supplements for all other purposes was not associated with having children under 18 years old (*p* > 0.05) (Table 6).

In 2021, unemployed respondents more frequently indicated the consumption of dietary supplements for boosting memory, digestion, and COVID-19 prevention (respectively, 9.7% vs. 16.7%, 15.5% vs. 21.1% and 4.6% vs. 7.9%; *p* < 0.05). The consumption of dietary supplements for all other purposes within the past 12 months was not associated with employment (*p* > 0.05). An increase in the consumption of dietary supplements appropriate for boosting energy, vision, the cardiovascular system, and COVID-19 prevention was also more frequently indicated by unemployed respondents (respectively, 20.9% vs. 27.8%, 19.9% vs. 27.3%, 33.8% vs. 41.9% and 20.9% vs. 27.8%; *p* < 0.05). An increase in the consumption of the exact dietary supplements for all other purposes during the pandemic was not associated with employment (*p* > 0.05) (Table 6).

During the pandemic, the consumption of dietary supplements for boosting memory and the nervous system was more frequently indicated by the respondents with lower income (respectively, 13.3% vs. 9% and 29.5% vs. 20.9%; *p* < 0.05). The consumption of dietary supplements for the cardiovascular system and digestion was more frequently indicated by the respondents with higher income (respectively, 23.1% vs. 29.9% and 12.1% vs. 17.9%; *p* < 0.05). The consumption of dietary supplements for other purposes was not associated with income (*p* > 0.05). Additionally, an increase in the consumption of the exact dietary supplements for all purposes during the pandemic was similar among respondents with lower and higher income (*p* > 0.05) (Table 6).

The consumption of dietary supplements for all purposes was similar regardless of the occurrence of COVID-19 among friends or relatives (*p* > 0.05). An increase in the consumption of dietary supplements appropriate for joints and bones was more frequently indicated among respondents who had suffered from COVID-19 personally or it was suffered by the respondent’s family members or friends (34.7% vs. 49.3%, *p* = 0.003). An increase in the consumption of the exact dietary supplements for all other purposes was not associated with personal suffering from COVID-19 or undergone by their family members or friends (*p* > 0.05) (Table 6).

The respondents who had suffered from COVID-19 or faced it in their families, more frequently indicated the consumption of dietary supplements for energy increase, as well as for COVID-19 prevention (respectively, 11.8% vs. 16.7% and 3.5% vs. 7.4%; *p* < 0.05). The consumption of dietary supplements for all other purposes was not associated with suffering from COVID-19 (*p* > 0.05). The respondents, who had COVID-19 personally or it occurred among their family members, more frequently indicated an increase in the consumption of dietary supplements appropriate for strengthening the immune system, general strengthening of the body, and joints and bones (respectively, 61.1% vs. 66.7%, 64.5% vs. 70.5% and 44.9% vs. 52.5%; *p* < 0.05). An increase in the consumption of dietary supplements appropriate for other purposes was not associated with suffering from COVID-19 by the respondents or their family members (*p* > 0.05) (Table 6).

The consumption of dietary supplements and an increase in the consumption of dietary supplements for all purposes was similar regardless the degree of the severeness of COVID-19 (*p* > 0.05) (Table 6).

## 4. Discussion

This study provided representative information about the consumption of dietary supplements among adult residents of Lithuania. The results of this study reveal that the consumption of dietary supplements was prevalent among the majority of the sample. Although the majority of the respondents indicated that the pandemic had had no effect on their use of dietary supplements, every fourth respondent’s consumption increased. Despite the differences observed in the consumption of dietary supplements among social and demographic groups, an increase in the consumption of dietary supplements during the pandemic was driven by the COVID-19 cases among the respondents or their family members or friends. Additionally, this study shows that most of the residents selected dietary supplements with an intention to strengthen the immune system or for the general strengthening of the body. An increase in the consumption of the exact dietary supplements appropriate for the general strengthening of the body and boosting the immune system was approximately twice more prevalent than consumption of dietary supplements affecting other parts of the human body. Additionally, a significant part of the sample indicated that they use dietary supplements with an intention to improve digestion. However, the prevalence of the increased consumption of the relevant dietary supplements during the pandemic was very low.

Surprisingly, this study revealed diverging results regarding the consumption of dietary supplements in 2019 when compared to the study conducted by other authors in Lithuania and included 2573 residents of Lithuania aged from 19 to 64 years. It revealed that the consumption of dietary supplements was prevalent among approximately 82% of adults. On the other hand, we could not unambiguously compare our results to the results of this study because of the different methods of data collection and analysis [15]. Our sample collected in 2019 revealed an increase in the consumption of dietary supplements. Additionally, according to the results of the statistical analysis of the data collected in 2021, every fourth respondent indicated that his or her consumption of dietary supplements increased during the pandemic. In addition, the consumption of dietary supplements increased among the respondents who had suffered from COVID-19 or their families or friends had suffered from this disease, as well as among those with a severe form of COVID-19. A study conducted in 2017 confirms that the prevalence of consumption of dietary supplements among adult residents of Lithuania significantly increased during the pandemic [16]. In 2017, only about half of adults (59.4%) used dietary supplements. The prevalence of the consumption was more often among females (67.7% vs. 50.2%), residents with higher education (64.9% vs. 48.8%), and persons with higher income. What is more, the most frequent reasons for consumption were strengthening of the immune system and disease prophylaxis (25.5% and 19.0%, respectively) [16]. With respect to gender and the degree of education, the results obtained are similar to the results of our study and in line with the study conducted in 2019. Interestingly, the results of our study did not reveal any difference in the prevalence of the consumption of dietary supplements with respect to the age groups. Previous studies conducted in Lithuania showed age to be an important factor for consumption of dietary supplements: the study conducted in 2019 revealed a higher prevalence of the consumption of dietary supplements among younger adults, the study of 2017 showed a higher prevalence among older adults [15,16]. Additionally, similar to our results, the study conducted in 2019 revealed that the consumption of dietary supplements was more prevalent among urban residents [15].

Studies conducted in other countries also showed similar results regarding an increase in the consumption of dietary supplements [6,17,18]. Additionally, the results show that, in all three countries, the consumption of dietary supplements for strengthening the immune system had increased [6,17,18]. Similarly, the study that was carried out in Spain and included more than a thousand adults revealed that the most frequently used supplements were combinations of multivitamins, minerals, and trace elements (27%), followed by vitamin D (25.8%) and vitamin C (22.2%) in variable doses [19]. These similarities and differences in the consumption of dietary supplements might be beneficial for the assessment of the knowledge gaps about various dietary supplements, and about behavioral differences related with cultural, regulatory, and other aspects.

A survey conducted in China in March 2020 also revealed that several dietary behaviors used to cope with COVID-19, including that an increased consumption of vitamin C, probiotics and other dietary supplements was associated with higher Household Dietary Diversity Scores. These scores were higher among older and urban residents, and among those with higher income. Gender, education level, family size, pregnancy, and having children under 5 years old in the household were not associated with the Household Dietary Diversity Score differences [18]. Surprisingly, our results show that the prevalence of consumption of dietary supplements was higher among respondents without children.

According to the studies on nutritional habits of adult residents of Lithuania, it is obvious that the majority of the adult population do not meet the nutritional recommendations [3,4], thus they are likely to face nutritional deficiencies [2,15]. The study conducted in 2019 [15] revealed that the daily consumption of cereal products and vegetables and fruits accounts for 49.5% and 57.1% of Lithuanian adult population, respectively. This leads to deficiencies of vitamins and minerals if no dietary supplements are taken. Our study provides valuable information about the prevalence of consumption of dietary supplements in the various social and demographic groups that could be used for targeted interventions in order to correct nutritional deficiencies and maintain an adequate intake of certain nutrients. According to the review performed by Harrison G.G. and published in 2010, targeted supplementation programs, enforced with other measures, may hold a great potential for a long-term progress regarding nutritional improvement, including micronutrient malnutrition. Specific targeted risk group programs were said to be crucial to both the individual and the population health in terms of food fortification. On the other hand, this applies to dietary supplementation as well. However, it was emphasized that it is important to take into account other constraining factors in order to ensure the success of targeted interventions. Despite the fact that there is some evidence that targeted educational approaches can work well, it is important to take into account factors such as unemployment and low income that cause significant barriers [20,21,22]. Our results showed similar tendencies.

Despite the fact that consumption of dietary supplements increased during the pandemic, a study, conducted in Poland revealed that the prevalence of consumption differed between the waves and, what is important, the prevalence in the third wave was not greater than in the first. During the third wave, the prevalence of the consumption of dietary supplements was likely to decrease, which indicates the necessity of additional studies on this topic in the near future [6].

### Limitations

A longitudinal study would have possibly provided more accurate results. On the other hand, this study included a representative sample larger than sufficient according to sex, age, place of residence, and education to reveal the consumption of dietary supplements among adult residents of Lithuania. Due to the data availability, we were not able to assess the representativeness according to other social and demographic factors.

Additionally, in order to assess the increase in the consumption of dietary supplements driven by specific purposes, we categorized dietary supplements with respect to the most impacted parts of the human body, despite the fact that dietary supplements are likely to have a much wider effect.

In addition, we did not collect the data about the time of consumption of dietary supplements with respect to the COVID-19 cases among the respondents or their families, friends.

Additionally, we analyzed only a limited number of possible factors that might have influenced the consumption of dietary supplements. In future studies, it would be beneficial to investigate this issue with more possible factors that might be related with the changes in the consumption of dietary supplements.

In order to assess the exact deficiencies in nutrients and the actual needs for the consumption of dietary supplements, it would be beneficial to investigate the actual nutrition taking into account the consumption of dietary supplements.

## 5. Conclusions

The consumption of dietary supplements was prevalent among the majority of adults in Lithuania and it increased during the COVID-19 pandemic. The consumption was more prevalent among females, respondents with university education, urban citizens, employed respondents, respondents without children, with higher income, and those who knew a person with COVID-19 infection. Most of the adults selected dietary supplements with an intention to strengthen the immune system or for the general strengthening of the body.

## Figures and Tables

**Figure 1 ijerph-19-09591-f001:**
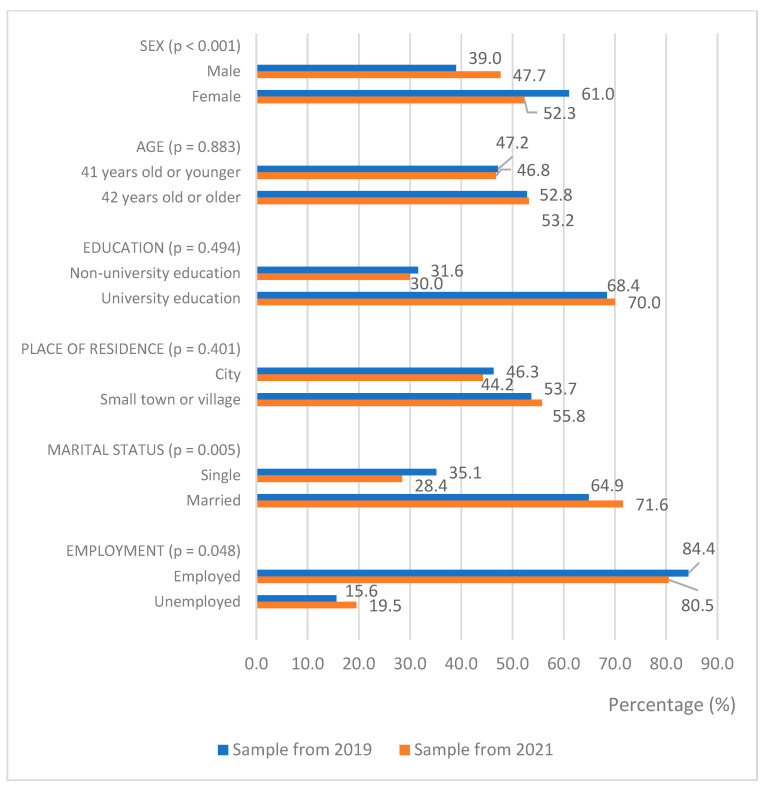
Distribution of the respondents who consumed dietary supplements by social and demographic characteristics in both samples.

**Table 1 ijerph-19-09591-t001:** Distribution of the respondents by social and demographic factors in both samples.

Factor	Sample from 2019			Sample from 2021	*p*-Value
	n	Relative Frequency (%)	N	Relative Frequency (%)	
**Sex**					
Male	402	46.2	792	49.5	0.118
Female	468	53.8	808	50.5	
**Age**					
41 years old or younger	421	48.4	769	48.1	0.876
42 years old or older	449	51.6	831	51.9	
**Education**					
Non-university education	229	26.4	474	31.9	0.005
University education	637	73.6	1010	68.1	
**Place of residence**					
City	378	43.4	678	42.4	0.607
Small town or village	492	56.6	922	57.6	
**Marital status**					
Single	298	34.7	377	28.1	0.001
Married	560	65.3	964	71.9	
**With children under 18 years old**					
No	-	-	991	61.9	-
Yes	-	-	609	38.1	
**Employment**					
Employed	714	83.1	1174	78.6	0.007
Unemployed	145	16.9	321	21.4	
**Income**					
EUR 350 or less	-	-	412	33.0	-
EUR 351 or more	-	-	836	67.0	
**COVID-19 among friends or relatives**					
Does not know anyone who suffered from COVID-19	-	-	184	11.5	-
The respondent or his/her family members or friends suffered from COVID-19	-	-	1416	88.5	
**COVID-19 among family members**					
There were no COVID-19 cases in the respondent’s family	-	-	970	60.6	-
The respondent or his/her family members suffered from COVID-19	-	-	630	39.4	
**Severeness of COVID-19**					
Suffered from asymptomatic or mild form of COVID-19	-	-	264	77.1	-
Suffered from a severe form of COVID-19	-	-	78	22.9	

**Table 2 ijerph-19-09591-t002:** Questions about the consumption of dietary supplements included in this study.

Question	Categories with Relevant Response Options *
Do you consume dietary supplements (vitamins, minerals, polyunsaturated fatty acids, plant-based preparations, etc.)?	Yes (yes, always/yes, more than 6 months per year/yes, 4–6 months per year/yes, 2–3 months per year/yes, 1 month per year/yes, but shortly or accidently) No (no, I do not consume) Excluded from the analysis (I do not know/cannot answer)
What dietary supplements and what for have you taken over the last 12 months? **	For strengthening the immune system/For disease prevention and the overall strengthening of the body/For energy boosting/For eye care/For boosting memory/For boosting the nervous system/For strengthening the cardiovascular system/For strengthening the joints, bones/For better digestion/For sleep regulation/For athletes/For weight regulation/For protection against the COVID-19 infection/Other
Has the COVID-19 pandemic made you more interested in dietary supplements?	Yes/No
How has the COVID-19 pandemic affected your habits of the consumption of dietary supplements?	Consumption of dietary supplements increased during the pandemic (I haven’t consumed them before, but I’ve started to consume them/I consumed them before, but my consumption increased)/Consumption of dietary supplements decreased or did not change during the pandemic (I consumed them before, but my consumption decreased/They had no effect)
Which of the dietary supplements were you made to consume more under the impact of the COVID-19 pandemic? **	Complex of vitamins and minerals/Complex of vitamins/Complex of minerals/Omega-3 fatty acids/Plant-based/Targeted at the immune system/Targeted at the cardiovascular system/Targeted at the nervous system/Targeted at the general strengthening of the body/Vitamin C/Vitamin D/Vitamins of the B group/Iron/Magnesium/Potassium/Calcium/Selenium/Coenzyme Q10/I do not know
Please select the appropriate statements for you: (Level of exposure to COVID-19) **	I’m suffering (or suffered) from COVID-19/There is (or was) a member in my family who is suffering (or suffered) from COVID-19/My friends, acquaintances, neighbours are suffering (or suffered) from COVID-19 in their families/I do not know anyone who is suffering (or suffered) from COVID-19
Please select the most appropriate statement about your COVID-19 infection:	Suffered from the asymptomatic or mild form of COVID-19 (I had an asymptomatic form of this disease/I had a mild form of this disease)/Suffered from a severe COVID-19 form (I had a severe form of this disease/I had a very severe form of this disease)

* In the case of larger categories, the response options are provided in brackets. ** Selection of multiple answer options available.

**Table 3 ijerph-19-09591-t003:** Dietary supplements and the purpose of their consumption.

	Consumption Purpose	For Strengthening the Immune System	For the Overall Strengthening of the Body	For Energy Boosting	For Vision	For Boosting Memory	For Boosting the Nervous System	For Strengthening the Cardiovascular System	For Strengthening the Joints and Bones	For Better Digestion	For Sleep Regulation	For Athletics	For Protection against COVID-19 Infection
Dietary Supplements													
Complex of vitamins and minerals		Yes	Yes	Yes	Yes	Yes	Yes	Yes	Yes		Yes	Yes	Yes
Complex of vitamins		Yes	Yes	Yes	Yes	Yes	Yes	Yes	Yes		Yes	Yes	Yes
Complex of minerals		Yes	Yes	Yes	Yes	Yes	Yes	Yes	Yes		Yes	Yes	Yes
Omega-3 fatty acids		Yes	Yes				No	Yes					Yes
Plant-based		Yes	Yes				Yes			Yes	Yes		Yes
Targeted at the immune system		Yes	Yes										Yes
Targeted at the cardiovascular system			Yes					Yes					
Targeted at the nervous system			Yes			Yes	Yes				Yes		
Targeted at the general strengthening of the body		Yes	Yes	Yes	Yes	Yes	Yes	Yes	Yes		Yes	Yes	Yes
Vitamin C		Yes	Yes										Yes
Vitamin D		Yes	Yes						Yes				Yes
Vitamins of the B group		Yes	Yes			Yes	Yes				Yes		Yes
Iron			Yes										
Magnesium			Yes				Yes				Yes		
Potassium			Yes					Yes					
Calcium			Yes						Yes				
Selenium		Yes	Yes					Yes					Yes
Co-enzyme Q10		Yes	Yes	Yes				Yes				Yes	Yes

Value “Yes” marks if a dietary supplement was assigned to the corresponding consumption purpose.

**Table 4 ijerph-19-09591-t004:** Distribution of the respondents by an increase in the consumption of dietary supplements during the COVID-19 pandemic by exact dietary supplements among the respondents who consumed dietary supplements (n = 1240).

Dietary Supplements	n	%
Complex of vitamins and minerals	93	7.5
Complex of vitamins	42	3.4
Complex of minerals	19	1.6
Omega-3 fatty acids	166	13.4
Plant-based	34	2.7
Targeted at the immune system	129	10.4
Targeted at the cardiovascular system	59	4.7
Targeted at the nervous system	45	3.6
Targeted at the general strengthening of the body	149	12.0
Vitamin C	381	30.7
Vitamin D	403	32.5
Vitamins of the B group	126	10.2
Iron	76	6.1
Magnesium	146	11.8
Potassium	61	4.9
Calcium	61	4.9
Selenium	46	3.7
Co-enzyme Q10	21	1.7
Other	38	3.1
Did not know	391	31.5

**Table 5 ijerph-19-09591-t005:** Distribution of the respondents by consumption of dietary supplements, the impact of the COVID-19 pandemic on the consumption of dietary supplements, sociodemographic factors, and COVID-19 cases.

Variable	Consumed Dietary Supplements in 2021		The Pandemic Triggered an Interest in Dietary Supplements		Consumption of Dietary Supplements Increased during the Pandemic	
	n (%)	*p*-Value	n (%)	*p*-Value	n (%)	*p*-Value
**Sex**						
Male	591 (75.5%)	0.012	173 (25.6%)	0.673	162 (27.4%)	0.297
Female	649 (80.7%)		188 (26.6%)		161 (24.8%)	
**Age**						
41 years old or younger	580 (76.3%)	0.095	154 (23.3%)	0.027	150 (25.9%)	0.876
42 years old or older	659 (79.8%)		206 (28.6%)		173 (26.3%)	
**Education**						
Non-university education	350 (75.1%)	0.006	100 (25.5%)	0.658	90 (25.8%)	0.704
University education	818 (81.3%)		241 (26.7%)		220 (26.9%)	
**Place of residence**						
City	548 (81.5%)	0.005	157 (26.8%)	0.608	154 (28.1%)	0.152
Small town or village	691 (75.6%)		203 (25.6%)		169 (24.5%)	
**Marital status**						
Single	294 (79.7%)	0.267	90 (28.2%)	0.299	87 (29.7%)	0.244
Married	740 (76.8%)		210 (25.2%)		193 (26.1%)	
**With children under 18 years old**						
No	783 (79.9%)	0.03	235 (27.5%)	0.133	206 (26.3%)	0.784
Yes	456 (75.2%)		126 (23.9%)		117 (25.6%)	
**Employment**						
Employed	936 (80%)	0.003	272 (26.3%)	0.694	253 (27.0%)	0.449
Unemployed	227 (72.3%)		72 (27.5%)		56 (24.6%)	
**Income**						
EUR 350 or less	307 (75.1%)	0.027	95 (27.4%)	0.934	80 (26.0%)	0.845
EUR 351 or more	670 (80.5%)		200 (27.1%)		178 (26.6%)	
**COVID-19 among friends or relatives**						
Does not know anyone who suffered from COVID-19	117 (65.7%)	<0.001	28 (20.3%)	0.103	19 (16.1%)	0.010
The respondent or his/her family members or friends suffered from COVID-19	1122 (79.7%)		332 (26.7%)		304 (27.1%)	
**COVID-19 among family members**						
There were no COVID-19 cases in the respondent’s family	738 (76.7%)	0.079	188 (22.7%)	<0.001	156 (21.1%)	<0.001
The respondent or his/her family members suffered from COVID-19	502 (80.4%)		173 (31.2%)		167 (33.3%)	
**Severeness of COVID-19**						
Had an asymptomatic or a mild form of COVID-19	200 (76.9%)	0.088	75 (32.3%)	0.145	65 (32.5%)	0.092
Had a severe form of COVID-19	67 (85.9%)		30 (41.7%)		29 (43.9%)	

The *p*-values mark the probability of an association between the social and demographic variables (including COVID-19 experiences) with the consumption of dietary supplements and changes in interests and their consumption. The first two pairs of columns represent the whole sample collected in 2021, excluding missing data (n = 1586); the last pair of columns represent the adults who consumed dietary supplements in the sample collected in 2021, excluding missing data (n = 1240).

**Table 6 ijerph-19-09591-t006:** Distribution of the respondents who consumed dietary supplements by the purpose of the consumption of dietary supplements and the consumption within past 12 months (in 2019 and 2021) and an increase in the consumption during the pandemic.

The Purpose of Consumption of Dietary Supplements	Consumed in 2019 (n = 585)		Consumed in 2021 (n = 1240)		Indicated an Increase in the Consumption of Relevant Dietary Supplements during the Pandemic (n = 1240)	
	N	%	n	%	n	%
Strengthening the immune system	189	32.3	608	49.0 *	785	63.3
The overall strengthening of the body	247	42.2	539	43.5	830	66.9
Energy boosting	129	22.1	171	13.8 *	271	21.9
Eye care	108	18.5	163	13.2 *	260	21.0
Boosting memory	82	14.0	134	10.8 *	359	29.0
Boosting the nervous system	115	19.7	290	23.4	431	34.8
Strengthening the cardiovascular system	127	21.7	334	26.9 *	434	35.0
Strengthening the joints, bones	80	13.7	311	25.1 *	594	47.9
Better digestion	80	13.7	202	16.3	34	2.7
Regulation of sleep	76	13.0	120	9.7 *	431	34.8
For athletics	53	9.1	69	5.5 *	271	21.9
Protection against COVID-19 infection	-	-	63	5.1	785	63.3

* Statistically significant difference in comparison with the sample collected in 2019 (*p* < 0.05).

## Data Availability

Not applicable.

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
