# Peer review of "COVID-19 Pandemic and Consumption of Dietary Supplements among Adult Residents of Lithuania"

_ijerph, 2022, doi:10.3390/ijerph19159591_

Round 1

Reviewer 1 Report

The authors present results from a survey sent to 1600 adults in Lithuania during the COVID pandemic, with the aim of studying use of supplements during the pandemic. The manuscrips has several serious flaws and limitations.  First, there is no comparator group, so the analysis rests entirely on self reports of supplement use during the pandemic. We don't know if in fact there was increased use of supplements during this time compared to another time period.  The survey instrument itself is redundant and it is difficult to understand how to interpret broad categories that likely overlap. For example, general consumption of a complex of vitamins and minerals might target all body systems. Additionally, there is no way to discern if there was a temporal relationship between taking supplements and COVID infection among respondents themselves, or among respondents' friends and family.  The discussion is a very lengthy restatement of the results, at times internally contradictory and difficult to follow. In Table 6, the categories seem as though they would  overlap--consumption over the past 12 mos likely includes the COVID pandemic. The data on supplements as tools for preventing COVID or mitigating disease severity are mixed, so statements in the introduction suggesting that there are public health interventions that might stem from these results are overstatements. The reviewer suggests submission to a different journal.

Author Response

Round 1.

Point 1: First, there is no comparator group, so the analysis rests entirely on self reports of supplement use during the pandemic. We don't know if in fact there was increased use of supplements during this time compared to another time period.

Response 1: Thank you for your review. In order to make objective comparisons with the pre-pandemic period, we used the data from our study performed in 2019, similar to the current one. Accordingly, the results section was updated.

Point 2: The survey instrument itself is redundant and it is difficult to understand how to interpret broad categories that likely overlap. For example, general consumption of a complex of vitamins and minerals might target all body systems.

Response 2: The Table 2 is presented in order to explain what food supplements were assigned to each purpose of their use. We chose such method presuming that people might not know what are the main targets of different food supplements. As shown in the Table 2, complex of vitamins and minerals was assigned to all body systems.

Point 3: Additionally, there is no way to discern if there was a temporal relationship between taking supplements and COVID infection among respondents themselves, or among respondents' friends and family. 

Response 3: We agree with your comment. Therefore, we updated the limitations section. On the other hand, we perforemd performed the statistical analysis and believe that it would be a very unlikely coincidence that there was other another factor but COVID-19 which lead to an increased consumption among respondents who suffered from COVID-19 by themselves or who had friends or family members with this disease.

Point 4: The discussion is a very lengthy restatement of the results, at times internally contradictory and difficult to follow.

Response 4: We updated the discussion and added some explanatory texts. However, we kept a large part of the restatements in order to emphasize the main results of our study.

Point 5: In Table 6, the categories seem as though they would  overlap--consumption over the past 12 mos likely includes the COVID pandemic.

Response 5: The main sample was formed and the data was collected in 2021. The question about the consumption of food supplements over the past 12 months includes only the pandemic period. The consumption within the past 12 months of the pandemic by the purposes of the consumption shows the opinion/selection criteria of food supplements of the respondents. The last pair of the columns shows the exact increase in consumption of food supplements groups calculated according to the methodology presented in Table 3 (former Table 2).

Point 6: The data on supplements as tools for preventing COVID or mitigating disease severity are mixed, so statements in the introduction suggesting that there are public health interventions that might stem from these results are overstatements.

Response 6: We deleted the statement about the public health interventions from the introduction.

Reviewer 2 Report

The manuscript aimed to assess the prevalence of the consumption of food supplements with respect to the purpose of consumption and sociodemographic factors among adults in Lithuania. The manuscript presents an interesting and pertinent theme. However, some reviews are necessary, mainly in the method that lacks information.

-       Abstract: The abstract should be a single paragraph and should follow the style of structured abstracts but without headings.

-       Line 14: Was the Search “before and during the pandemic”? It is different from line 59.

-       Introduction: explore better the benefits (or not) related to supplement consumption. What kind of supplement is your target?

-       Method: The study design and method used to access the period before the pandemic is unclear.

-       Lines 61-64: How did you control the sample to be representative using an online survey?

-       Line 77 – were the new questions (related to COVID-10) previously validated?

-       Lines 102 -103: What kind of supplements?

-       Table 3 and others – change “N” to “n”.

-       Why does Table 4 present 1381 respondents?

-       It is unclear in the manuscript if the sample used in the previous study is the same.

-        Line 364: was it representative regarding income, educational level, etc.?

Thank you for the opportunity to review this manuscript!

Author Response

Round 1.

Point 1: Abstract: The abstract should be a single paragraph and should follow the style of structured abstracts but without headings.

Response 1: Thank you for your review. We deleted the headings from the abstract.

Point 2: Line 14: Was the Search “before and during the pandemic”? It is different from line 59.

Response 2: We added an additional data collected in 2019 to this study. Accordingly, it is true that we collected data about the consumption of food supplements before and during the pandemic. On the other hand, in this paper, we focused on the consumption during the pandemic.

Point 3: Introduction: explore better the benefits (or not) related to supplement consumption. What kind of supplement is your target?

Response 3: The aim of our paper was to present the change in the consumption of food supplements in various social and demographic groups of Lithuanian residents triggered by the COVID-19 pandemic. We aimed to reveal the general view of the consumption of a complete spectrum of food supplements.

Point 4: Method: The study design and method used to access the period before the pandemic is unclear.

Response 4: We added an additional sample for assessment of the consumption of food supplements during the pre-pandemic period. Accordingly, we updated the Methods section in order to explain this.

Point 5: Lines 61-64: How did you control the sample to be representative using an online survey?

Response 5: We updated the Methods section and provided the explanation.

Point 6: Line 77 – were the new questions (related to COVID-10) previously validated?

Response 6: We updated the Methods section and provided an answer to this question.

Point 7: Lines 102 -103: What kind of supplements?

Response 7: We added a reference to the Table 2 (former Table 1).

Point 8: Table 3 and others – change “N” to “n”.

Response 8: We changed “N” to “n” in all tables.

Point 9: Why does Table 4 present 1381 respondents?

Response 9: We updated the heading of the table and added the explaination.

Point 10: It is unclear in the manuscript if the sample used in the previous study is the same.

Response 10: We added an explanatory phrase “performed by other authors” in order to clarify this. All studies, cited in the discussion, included samples other than described in the Methods section.

Point 11: Line 364: was it representative regarding income, educational level, etc.?

Response 11: We updated the limitations section and provided an answer to this question.

Reviewer 3 Report

1.      Line 20-22  – this sentence is unclear. What are the two groups that have these data- “The consumption of relevant food supplements increased among 63.3% and 66.9% of respondents, respectively”. In the previous sentence there are also two values with a respectively that seem to refer to % for immunity and % for body health.

2.      Line 30 – please define “food supplements.” It may be unclear as in different countries this might be perceived differently.  Many researchers prefer “dietary supplements” or “nutritional supplements” as using the word “food” implies a food--like substance. Are these pills, tablets etc. or actual foods to supplement the diet (liquid or bars) or all of these variations?

3.      Line 60 – what was the minimum age?

4.      Table 2. In a footnote please summarize what a “yes” answer means.

5.      Paragraph beginning line 296. Here we are told prevalence have not changed since 2019 yet all through the paragraph are statements describing an increase in use of supplements. This is unclear.  It seems that frequency of use is the main factor to discuss so these data should be shown to compare prepandemic and post-pandemic data.

6.      Lines 306-309 add to the confusion as it states: “Another previous study confirms that the prevalence of consumption of food supplements among adult residents of Lithuania significantly increased. In 2017, only about half of adults (59.4%) … “ This needs to be reworded. If you have data going back in time please present so the reader can see changes over time. What happened between 2017 and 2019 is not really of interest however if there has been a study increase over time that peaked in 2019 this should be described.  Further it seems there are large age differences in these 2019 and 2017 studies that have distorted findings to make less comparable.  

7.      Line 325 – why compare to Saudi Arabia? It is a very different country. Generally it is not if interest to compare to different countries without acknowledging cultural differences and there may be different policies and regulations in the countries.

8.       Line 354- you cite references 19-21 as being yours yet these studies are done by others.

9.      You have many more limitations than  what you acknowledge. These need to be addressed.

10.   Line 373 – here you say supplement use increased yet do not have evidence for this as you point out in line 296.

Author Response

Round 1.

Point 1: Line 20-22  – this sentence is unclear. What are the two groups that have these data- “The consumption of relevant food supplements increased among 63.3% and 66.9% of respondents, respectively”. In the previous sentence there are also two values with a respectively that seem to refer to % for immunity and % for body health.

Response 1: Thank you for your review. We rephrased the mentioned sentence in order to clarify this.

Point 2: Line 30 – please define “food supplements.” It may be unclear as in different countries this might be perceived differently.  Many researchers prefer “dietary supplements” or “nutritional supplements” as using the word “food” implies a food--like substance. Are these pills, tablets etc. or actual foods to supplement the diet (liquid or bars) or all of these variations?

Response 2: We agree that it was not completely clear what we meant by “food supplements”, therefore we replaced “food supplements” with the definition “dietary supplements”.

Point 3: Line 60 – what was the minimum age?

Response 3: We updated the Methods section and added the minimum age of the respondents.

Point 4: Table 2. In a footnote please summarize what a “yes” answer means.

Response 4: We added a footnote with an explanation.

Point 5: Paragraph beginning line 296. Here we are told prevalence have not changed since 2019 yet all through the paragraph are statements describing an increase in use of supplements. This is unclear.  It seems that frequency of use is the main factor to discuss so these data should be shown to compare prepandemic and post-pandemic data.

Response 5: We added an additional sample to our study in order to compare the consumption of dietary supplements during the pre-pandemic period with the consumption of dietary supplements during the pandemic. Accordingly, we updated the discussion in order to clarify this.

Point 6: Lines 306-309 add to the confusion as it states: “Another previous study confirms that the prevalence of consumption of food supplements among adult residents of Lithuania significantly increased. In 2017, only about half of adults (59.4%) … “ This needs to be reworded. If you have data going back in time please present so the reader can see changes over time. What happened between 2017 and 2019 is not really of interest however if there has been a study increase over time that peaked in 2019 this should be described.  Further it seems there are large age differences in these 2019 and 2017 studies that have distorted findings to make less comparable.

Response 6: We reworded the mentioned sentence.

Point 7: Line 325 – why compare to Saudi Arabia? It is a very different country. Generally it is not if interest to compare to different countries without acknowledging cultural differences and there may be different policies and regulations in the countries.

Response 7: We removed the comparison of the consumption of dietary supplements between Lithuania and Saudi Arabia but kept the statement that even in Saudi Arabia the consumption of dietary supplements has increased during the pandemic.

Point 8: Line 354- you cite references 19-21 as being yours yet these studies are done by others.

Response 8: In order to clarify, we moved the comparison with the results of our study to a separate sentence.

Point 9: You have many more limitations than  what you acknowledge. These need to be addressed.

Response 9: We updated the limitations section.

Point 10: Line 373 – here you say supplement use increased yet do not have evidence for this as you point out in line 296.

Response 10: We added a sample collected in 2019 to our analysis and it confirmed our conclusion.

Reviewer 4 Report

The aim of this paper was to investigate changes in nutritional supplements during the Covid-19 pandemic.  It is novel as it is based in Lithuania, although there has been previous research on this in other countries.

Overall this paper will be of some interest to the reader and is insightful to changes in supplement intake during the Covid pandemic.  However, there are some areas that need addressing before the paper can be published and I have set these out below:-

In the introduction you need to justify why you have stratified your data for 7 years, 18 years and 42 years.  As there is so much data in this paper I would be inclined to just use below 18y and 18y and above, as this would determine adults from children.  I don't see the point of 7 year or 42 year, unless this has a meaning in Lithunaia, in which case this needs to be stated.

You have categorised the supplements into different effects on the body, for example for vision.  However, I am unsure what 'general strengthening of the body is'. All of the supplements appear to answer 'yes' to this, but I think it is too generic. For example how would potassium help with general strenthening of the body?  I would either remove this category or explain what it means in the introduction.

Table 3, should be in the methods as it is your participant characteristics not results

Table 4, in the final column, you have stated 'Relative Frequency (%)'. I believe this to just be % and I would remove the words 'relative frequency'.  This would be for example 99/1381, which is the same as the %, so just say percentage.  As this comes to more than 100%, I presume that this is because some people took more than 1 supplement, this needs clarifying.  It would also be good to arrange this table in descending order to show the increase more clearly.

Table 5 requires a column that says 'Does not consume food supplements' this would make it easier for the reader than having to deduct the percentages given from 100 to find out how many didn't take the supplements. I think this would be of interest to the reader, too.  I am also not sure what the P value represents in table 5.  Is it the difference between each variable eg male vs female?  For example for the sex variable what does p=0.012 actually mean?  This needs clarifying.

For table 6 it would be better to show percentage change rather than absolute percentage/relative frequency as it is the change that is really the research question, not the absolute values.

Line 289 about general strengthening, is a little difficult to understand and could be re-worded.

Line 307 onwards, you have frequently stated 'another study' but you need to state which study you are referring to by giving the author and year, even if you are then referencing it with a number. You need more references throughout this whole paragraph.

You have gone on to compare previous studies in Poland, Saudi and China, stating that intake in Lithunania is higher or lower than the other countries.  But what does this actually tell us?  is this a good thing or a bad thing?  What does this information  mean to the people or the country or to the supplement industry?  It needs a bit more explaining, otherwise is just a list of which countries have a higher or lower intake.

Line 341 - you can't make the assumption of nutritional deficiencies as you have not recorded food intake.

Line 343 - Harrison G.G. needs the year.

Overall, there is a lot of information in this paper and it is quite hard to see what the message is. It would be improved if some of the categories were removed so that the main focus of the research question - whether the supplements increased during lockdown- would be easier to see. You may wish to consider removing the data for the 7y and 42 year sub sections as it doesn't have much relevance.  Also, maybe the number of family members as no significance is shown there.  This would just help the reader to focus on the more relevant information.  There is quite a lot of information in the results section and some of this could be better presented in graphs, which would be easier to follow.

Author Response

Round 1.

Point 1: In the introduction you need to justify why you have stratified your data for 7 years, 18 years and 42 years.  As there is so much data in this paper I would be inclined to just use below 18y and 18y and above, as this would determine adults from children.  I don't see the point of 7 year or 42 year, unless this has a meaning in Lithunaia, in which case this needs to be stated.

Response 1: Thank you for your review. In order to clarify the presentation of the results of our study we removed a variable about having children under 7 years old and kept only a variable of having children under 18 years old.

Point 2: You have categorised the supplements into different effects on the body, for example for vision.  However, I am unsure what 'general strengthening of the body is'. All of the supplements appear to answer 'yes' to this, but I think it is too generic. For example how would potassium help with general strenthening of the body?  I would either remove this category or explain what it means in the introduction.

Response 2: We added an explanation of the categorization. This category was added because of the possible consumption of dietary supplements without a specific target in the human body.

Point 3: Table 3, should be in the methods as it is your participant characteristics not results

Response 3: We moved the Table 3 to the Methods section.

Point 4: Table 4, in the final column, you have stated 'Relative Frequency (%)'. I believe this to just be % and I would remove the words 'relative frequency'.  This would be for example 99/1381, which is the same as the %, so just say percentage.  As this comes to more than 100%, I presume that this is because some people took more than 1 supplement, this needs clarifying.  It would also be good to arrange this table in descending order to show the increase more clearly.

Response 4: We removed phrases “Relative frequency” from the tables. Also, we added an explanation to the Methods section about questions where the selection of multiple answer options was available.

Point 5: Table 5 requires a column that says 'Does not consume food supplements' this would make it easier for the reader than having to deduct the percentages given from 100 to find out how many didn't take the supplements. I think this would be of interest to the reader, too.  I am also not sure what the P value represents in table 5.  Is it the difference between each variable eg male vs female?  For example for the sex variable what does p=0.012 actually mean?  This needs clarifying.

Response 5: Thank you for the suggestion regarding the numbers of respondents who did not consume dietary supplements. However, because of the technical limitations (width of the page) we could not add an additional column. According to the fact that readers may deduct percentages from 100 in order to find the percentage of those who did not consume dietary supplements, we believe that the table is informative enough to satisfy the mentioned interest. On the other hand, we added a footnote with an explanatory text about meaning of the p-values.

Point 6: For table 6 it would be better to show percentage change rather than absolute percentage/relative frequency as it is the change that is really the research question, not the absolute values.

Response 6: We added an additional group of columns representing the consumption of dietary supplements in 2019. However, we believe that it is more informative to provide exact percentages of the prevalence of consumption rather than presenting changes in those percentages.

Point 7: Line 289 about general strengthening, is a little difficult to understand and could be re-worded.

Response 7: We re-worded the sentence.

Point 8: Line 307 onwards, you have frequently stated 'another study' but you need to state which study you are referring to by giving the author and year, even if you are then referencing it with a number. You need more references throughout this whole paragraph.

Response 8: We added a reference and re-worded the mentioned sentence.

Point 9: You have gone on to compare previous studies in Poland, Saudi and China, stating that intake in Lithunania is higher or lower than the other countries.  But what does this actually tell us?  is this a good thing or a bad thing?  What does this information  mean to the people or the country or to the supplement industry?  It needs a bit more explaining, otherwise is just a list of which countries have a higher or lower intake.

Response 9: We added an explanaroy explanatory sentence to the manuscript.

Point 10: Line 341 - you can't make the assumption of nutritional deficiencies as you have not recorded food intake.

Response 10: We added an explanatory sentence to the discussion. Also, we updated the limitations section.

Point 11: Line 343 - Harrison G.G. needs the year.

Response 11: We added the year near the reference.

Point 12: Overall, there is a lot of information in this paper and it is quite hard to see what the message is. It would be improved if some of the categories were removed so that the main focus of the research question - whether the supplements increased during lockdown- would be easier to see. You may wish to consider removing the data for the 7y and 42 year sub sections as it doesn't have much relevance.  Also, maybe the number of family members as no significance is shown there.  This would just help the reader to focus on the more relevant information.  There is quite a lot of information in the results section and some of this could be better presented in graphs, which would be easier to follow.

Response 12: We removed analysis according the numver of family members and having children under 7 years old.

Round 2

Reviewer 1 Report

The addition of a comparison group from 2019 strengthens this manuscript. There are a few additional revisions that this reviewer requests:

1. Table 1 should include p values. There are p values discussed for this table discussed in the results section, however they should be in the table itself

2. The results section is an overly lengthy summary of the data, without clear focus on the key findings

3. You state on page 14, new text, lines 372-373, that a large part of the Lithuanian population faces nutritional deficiencies. Reference #3 does not support that. The reviewer was unable to access the other references for this claim. Please elaborate on what deficiencies are prevalent in the Lithuanian population? This part of the discussion seems out of context since the manuscript is focusing on the increase in dietary supplements specifically during COVID, not on consumption of dietary supplements to correct nutritional deficiencies. 

Author Response

Round 2.

Point 1: Table 1 should include p values. There are p values discussed for this table discussed in the results section, however they should be in the table itself

Response 1: Thank you for your review. We added the p-values.

Point 2: The results section is an overly lengthy summary of the data, without clear focus on the key findings

Response 2: We agree with your comment. However, we believe that our results provided might be interesting for various readers. Therefore, we did not deleted any part of our results but rearranged them and added the subsections in order to make the results section more –reader friendly.

Point 3: You state on page 14, new text, lines 372-373, that a large part of the Lithuanian population faces nutritional deficiencies. Reference #3 does not support that. The reviewer was unable to access the other references for this claim. Please elaborate on what deficiencies are prevalent in the Lithuanian population? This part of the discussion seems out of context since the manuscript is focusing on the increase in dietary supplements specifically during COVID, not on consumption of dietary supplements to correct nutritional deficiencies.

Response 3: In Round 1, we added the comment about nutritional deficiencies according to the comments of Reviewer 4. At this time we re-worded the mentioned sentence and added an additional comment about nutritional deficiencies with respect to the nutritional habits of Lithuanian population of adults.

Reviewer 2 Report

Dear authors, based on your responses and adjustments in the manuscript, I am concerned with some points highlighted below.

1) Lines 66 - -69: When an individual refused to participate, how did you select another one?

2) Considering that you did not use the same sample before and during the pandemic, you should use the comparison only in the discussion section. I recommend you remove the previous study from the method and result section and focus only on the consumption of dietary supplements only during the pandemic (comparing it in the discussion section).

3) Table 1 - Since the 2019 data is from another study, it should be referenced.

4) Lines 88 - 97: Were the questions previously validated?

Thank you for the opportunity to review this manuscript.

Author Response

Round 2.

Point 1: Lines 66 - -69: When an individual refused to participate, how did you select another one?

Response 1: Thank you for your review. We added an explanation to the Methods section.

Point 2: Considering that you did not use the same sample before and during the pandemic, you should use the comparison only in the discussion section. I recommend you remove the previous study from the method and result section and focus only on the consumption of dietary supplements only during the pandemic (comparing it in the discussion section).

Response 2: In Round 1, we added an additional sample with respect to the comments of the Reviewer 1. In Round 2, Reviewer 1 responded stating that “the addition of a comparison group from 2019 strengthens this manuscript”. We also believe that addition of the sample collected in 2019 does not decrease the quality of our study and serves as an evidence that the consumption of dieatary supplements trully increased during the pandemic and it is not based only on subjective views of the respondents. What is more, both samples were similar according to sex, age and place of residence. Also, we believe that the appropriate place for presentation of the results of the 2019 sample is the Results section because we used an actual data collected in 2019 and not the publication presenting a summary of such results which could be discussed.

Point 3: Table 1 - Since the 2019 data is from another study, it should be referenced.

Response 3: The data collected in 2019 belongs to the same study. It was the pilot stage of our study conducted in 2021. We have the actual data collected in 2019 in our dataset.

Point 4: Lines 88 - 97: Were the questions previously validated?

Response 4: Two anonymous validated original questionnaires were used for this study. This is mentioned in the Methods section.